# The effect of melatonin supplementation on the plasma levels of 2-arachidonoylglycerol, ghrelin and hedonic eating intensity in overweight/obese females: A study protocol for a pilot randomized controlled trial

**Malihe Karamizadeh[1,2], Azadeh Khalilitehrani[1,2], Neda Lotfi Yagin[3], Marzieh Akbarzadeh[4], Reza Mahdavi[5]\*, Bahram Pourghassem Gargari [5]\***

**1** Student Research Committee, Tabriz University of Medical Sciences, Tabriz, Iran, **2** Faculty of Nutrition and Food Sciences, Tabriz University of Medical Sciences, Tabriz, Iran, **3** Endocrine Research Center, Tabriz University of Medical Sciences, Tabriz, Iran, **4** Nutrition Research Center, School of Nutrition and Food Sciences, Shiraz University of Medical Sciences, Shiraz, Iran, **5** Department of Biochemistry and Diet Therapy, Faculty of Nutrition and Food Sciences, Nutrition Research Center, Tabriz University of Medical Sciences, Tabriz, Iran

\* mahdavir@tbzmed.ac.ir (RM); pourghassemb@tbzmed.ac.ir (BPG)

## Abstract

### Introduction

Hedonic eating, reward-driven eating rather than out of biological needs, has been proposed as one of the important causes of overweight and obesity in recent years. Dopamine, endocannabinoids, opioids, and ghrelin are among the physiological factors associated with hedonic eating. Since the results of some previous animal studies have indicated the effectiveness of melatonin supplementation on the levels of endocannabinoids, and ghrelin, therefore this pilot study will investigate the effect of melatonin supplementation on plasma levels of endocannabinoid 2-arachidonylglycerol, ghrelin, and the intensity of hedonic eating in overweight/obese females.

### Methods

In a randomized, double-blinded, placebo-controlled study, forty-six women with overweight/obesity and high hedonic eating intensity (total score of power of food scale > 2.5) will be recruited. They will receive either a 5 mg/day melatonin supplement (n = 23) or a placebo (n = 23) for 8 weeks. The primary outcomes, including the plasma levels of 2-arachidonylglycerol and ghrelin, and the intensity of hedonic eating will be assessed at the baseline and end of the study. Additionally, the secondary outcomes (dietary intake, and body weight) will be evaluated at the study's onset, after four weeks, and upon completion of the intervention. A one-way analysis of covariance (ANCOVA) will be used to detect the effect of melatonin supplementation on outcome variables.

**Data availability statement:** This study is a protocol study, and no datasets were generated or analyzed during the current study.

**Funding:** The funding for the trial received support from Tabriz University of Medical Sciences under grant number IR.TBZMED.REC.1402.075. The funders had no role in study design, data collection and analysis, decision to publish, or preparation of the manuscript.

**Competing interests:** The authors have declared that no competing interests exist.

## Discussion

Considering the positive effects of melatonin supplementation in reducing endocannabinoid levels, the expression of the ghrelin hormone gene, the level of ghrelin, and the cannabinoid receptor type 1 gene expression in animal studies, it is possible that in human subjects, it could impact the intensity of hedonic eating by lowering endocannabinoid and ghrelin levels.

## Trial registration

The trial was registered with the Iranian Registry of Clinical Trials in June 2023 under the ID number IRCT20080904001197N22.

## Introduction

Appetite and food intake are complex functions that are partially controlled by two systems: the homeostatic system and the hedonic system. The homeostatic system responds to biological signals related to energy levels, such as depleted or replenished energy stores, while the hedonic system is associated with reward mechanisms related to dopamine. Especially palatable foods activate reward mechanisms, leading individuals to consume them repeatedly. Over time, these processes can contribute to weight gain and obesity [1].

In hedonic eating, palatable foods can activate the brain's reward circuits by releasing dopamine. Additionally, these types of foods can lead to the release of endocannabinoids and opioids, which in turn may help increase the release of dopamine [2–4].

Endocannabinoids are a crucial component of the endocannabinoid system, which includes of cannabinoid receptors, such as cannabinoid receptor type 1 (CB1) and cannabinoid receptor type 2 (CB2), as well as their endogenous ligands, like 2-arachidonylglycerol (2-AG) and N-arachidonylethanolamine (AEA). This system plays an important role in regulating food intake and hedonic eating. In the limbic system, activation of CB1 receptors by endocannabinoids released during the eating of palatable foods can increase food intake especially hedonic eating by stimulating the release of dopamine [4–8]. Also, in addition to ghrelin's role in the homeostatic regulation of food intake, previous studies have shown that ghrelin level rises rather than drop in response to the consumption of palatable foods under satiety conditions. It has been proposed that ghrelin plays a significant role in hedonic eating by stimulating the release of dopamine, increasing the activity of dopamine receptors, and affecting the opioid receptors involved in food reward [9–11]. In the study conducted by Monteleone et al., consuming food for pleasure increased the plasma levels of ghrelin and 2-AG, and the levels of these two substances showed a positive correlation [2].

In addition to biochemical parameters, such as endocannabinoids and ghrelin, the Power of Food Scale (PFS), designed in 2009 by Lowe et al., is routinely used to assess individual differences in hedonic eating intensity [12]. The PFS is a self-report instrument used to assess individual differences in thoughts, feelings, and motivations related to appetite in environments where palatable foods are available [13].

Due to the important role of hedonic eating in the development of overweight and obesity, weight management interventions targeting hedonic eating have been linked to greater weight loss [14–16]. Melatonin, a hormone secreted from the pineal gland, has several roles, including regulating sleep and circadian rhythm, promoting weight loss, and exhibiting antioxidant and anti-inflammatory properties [17–19]. Animal studies have shown positive effects of melatonin supplementation on reducing levels of endocannabinoids, ghrelin, and CB1 expression

[7,20–24], suggesting it may also be effective in hedonic eating. To date, no published study has specifically investigated the effect of melatonin supplementation on hedonic eating. Therefore, this pilot study will investigate the effects of melatonin supplementation on plasma levels of 2-AG, ghrelin, and the intensity of hedonic eating in women who are overweight or obese and have high hedonic eating intensity.

### Hypothesis

We hypothesized that melatonin supplementation may improve hedonic eating intensity by affecting levels of endocannabinoid 2-AG, and ghrelin.

## Methods and design

### Study design

The present pilot study is a two-arm, parallel-group, randomized, double-blind, and placebo-controlled superiority clinical trial. The timeline and flowchart of the study are presented in Figs 1 and 2. This intervention will take place in the Nutrition Research Laboratory of the Faculty of Nutrition and Food Sciences at Tabriz University of Medical Sciences, Tabriz, Iran. Its purpose is to investigate the effects of melatonin supplementation on the plasma concentrations of 2-AG and ghrelin, as well as on the hedonic eating score, dietary intake, and body weight in women with overweight/obesity and high hedonic eating intensity. The trial's protocol adheres to the recommendations for a clinical trial protocol (S1 Checklist, SPIRIT 2013 checklist).

### Sample size

Currently, no human studies have investigated the use of melatonin supplementation on hedonic eating. The proposed study is a pilot project; its main aim is to establish data for sample size calculations for a larger trial. Various studies recommend a total sample size of at least 24 participants for a two-arm trial [25]. Considering the sample size of previous studies on melatonin supplementation in overweight and obese Iranian women [26,27] and to increase the accuracy and power of the study, in this clinical trial, we plan to enroll 40 participants, which, with a 15% attrition rate, will increase to 46 subjects (23 in each group).

### Study population

The present pilot study is the second phase of a Ph.D. dissertation. The first phase of the project is cross-sectional, and approximately 400 overweight or obese females (BMI: 25–34.9 kg/m²) will be recruited through advertisements in public places, social media, fitness centers, etc. The high hedonic eating intensity will be assessed using the PFS, which has previously been validated in the Persian population [28]. Forty-six volunteers who exceed the threshold score, meaning a total PFS score > 2.5 [29,30], will proceed to the next phase of the research, which involves intervention. All eligible participants will receive comprehensive information regarding the research procedures by the principal investigator, and consent will be obtained by signing an informed consent document (S1 File).

### Inclusion and exclusion criteria

For the current study, 46 adult women meeting the following inclusion criteria will be recruited: apparently healthy volunteers (individuals who have not been diagnosed with any medical conditions and are not taking any medications), literate, aged 19–49 years; BMI = 25–34.9 kg/m²; exhibiting high hedonic eating intensity (total PFS score > 2.5); and expressing a desire to use drugs or supplements for weight loss instead of dieting.

| | STUDY PERIOD | | | | | | | | | | | | |
|---|---|---|---|---|---|---|---|---|---|---|---|---|---|
| | Enrolment | Allocation | Post-allocation weeks | | | | | | | | | | Close-out |
| TIMEPOINT** | $-t_1$ | 0 | Baseline | 1 | 2 | 3 | 4 | 5 | 6 | 7 | 8 | | Week 9 |
| **ENROLMENT:** | | | | | | | | | | | | | |
| *Eligibility screen* | X | | | | | | | | | | | | |
| *Informed consent* | X | | | | | | | | | | | | |
| *Randomization* | | X | | | | | | | | | | | |
| *Allocation* | | X | | | | | | | | | | | |
| *Participant training* | | | X | | | | | | | | | | |
| **INTERVENTIONS:** | | | | | | | | | | | | | |
| *Supplementation* | | | | | | | | | | | | | |
| *Compliance* | | | | | | | | | | | | | |
| *Adverse events* | | | | | | | | | | | | | |
| **ASSESSMENTS:** | | | | | | | | | | | | | |
| *Hedonic eating intensity* | X | | | | | | | | | | | | X |
| *Biochemical biomarkers* | | | X | | | | | | | | | | X |
| *Dietary intake* | | | X | | | | X | | | | | | X |
| *Body weight* | | | X | | | | X | | | | | | X |
| *Physical activity* | | | X | | | | | | | | | | X |
| *Supplement monitoring* | | | | X | X | X | X | X | X | X | X | | |

**Fig 1. SPIRIT Schedule of enrolment, intervention, and assessments of the outcomes.**

Participants will be excluded based on the following criteria: being postmenopausal, pregnant, or lactating; having irregular menstruation; recent weight loss or participation in weight loss programs; current smoking status; recent use of supplements affecting appetite or weight (within the last 3 months); substance abuse; alcohol consumption; any chronic diseases; use of appetite-affecting drugs (such as antidepressants, steroids, and oral contraceptives); working night shifts; severe insomnia [as determined by the Insomnia Severity Index (ISI)]; use of drugs that interfere with melatonin (such as anticonvulsant drugs, anticoagulant drugs, sedatives).

As mentioned, the present pilot study is the second phase of a Ph.D. dissertation. The project's first phase is cross-sectional, and approximately 400 overweight or obese females will be recruited. The mentioned inclusion and exclusion criteria will be assessed in the first phase of the project.

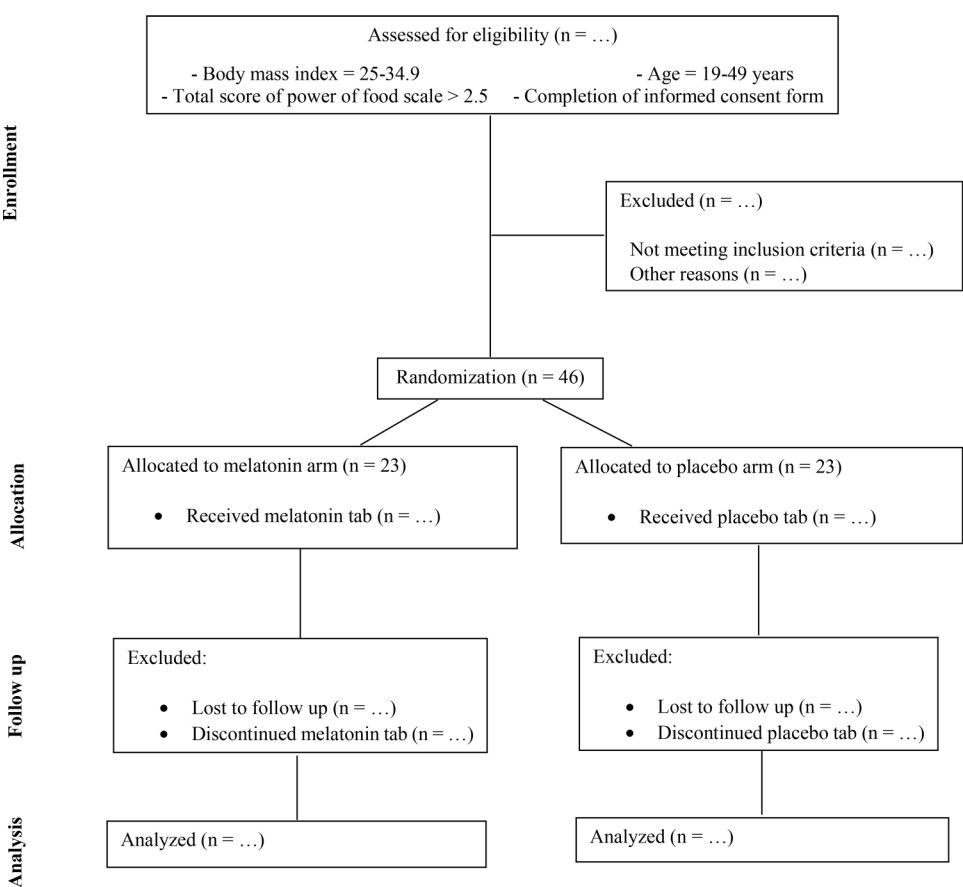

**Fig 2. Study protocol flowchart, adapted from CONSORT.**

## Randomization, sequence generation, and concealment

Participants will be randomly allocated into two groups: one receiving melatonin tablets and the other receiving placebo tablets (a combination of silicon dioxide, cellulose, and starch). The stratified blocked randomization method will be employed to assign individuals to the two groups. The sequence will be generated by one of the authors (RM), who will not be involved in the participant assignment. Allocation concealment is ensured, as the author will not disclose the randomization code until the participant has been recruited into the trial by other authors (MK and AKh), following the completion of all baseline measurements. Participants are categorized as overweight (BMI: 25–29.9 kg/m²) or obese (BMI: 30–34.9 kg/m²) and are then randomly assigned to either the melatonin or placebo group. Randomization is conducted separately within each BMI category using the Random Allocation Software (RAS). The block size is set at four, with two allocations for the melatonin group (M) and two for the placebo group (P).

## Blinding

The present pilot study is designed as a double-blind trial, ensuring that neither the researchers conducting the study nor the participants will know which group they have been assigned to. To maintain the double-blind nature, the packages containing melatonin and placebo tablets will be coded as 'A' and 'B' at the start of the study by an individual not involved in the

research process. This coding serves to blind the researchers. Furthermore, the supplement and placebo pills will be made identical in color, shape, smell, and appearance to blind the participants.

Labels on the supplement packs will stay hidden until the study is finished. However, if any adverse events occur, unblinding is allowed according to the Medical Ethics Committee's guidelines to determine whether the patient received melatonin or a placebo. If dropout rates reach 15% or higher due to adverse effects, we will conduct an intention-to-treat (ITT) analysis.

## Implementation

MK and AKh will send out invitations to potential participants for the study. Meanwhile, RM will be responsible for creating the sequence but will not be involved in the assignment of participants to the study. RM is also blinded to any participant details or baseline data.

## Intervention

Forty-six eligible women will be randomly assigned to either the melatonin group or the placebo group, with 23 participants in each group. Participants will take a 5 mg tablet one hour before bedtime with sufficient water and will be followed up for 8 weeks. The Tehran Darou Pharma & Food Supplement Company will supply the melatonin and placebo tablets, which will be identical in weight, size, shape, taste, color, odor, and packaging. Patients will receive the tablets during monthly study visits, with a total of three visits scheduled for the trial (before the intervention, after 1 month of intervention, and at the end of the intervention). Patients will be instructed on how to use their tablets and will receive weekly follow-up calls to improve response rates and fewer dropouts.

To enhance participant retention, the potential health benefits of melatonin will be explained, and the principal investigator will keep in regular contact with participants via phone calls.

Compliance will be assessed based on the number of unused tablets returned by each participant at the end of week four and the end of the trial. The remaining tablets will be counted to determine the total intake of the tablets. The total intake of tablets for each subject is calculated by subtracting the number of tablets unused from the number of tablets dispensed.

Participants can choose to leave the study at any point. Furthermore, the investigator has the authority to remove subjects from the study if they do not adhere to the protocol. Additionally, patients will be monitored for any adverse events following melatonin consumption. If any adverse events are attributed to melatonin, participants will be advised to discontinue the supplements and will be promptly referred to a specialist for further treatment.

It is important to note that no specific physical activity program, dietary recommendations, or regimens will be provided to the study subjects. Additionally, after considering the eligibility criteria, if subjects are being treated for any kind of disease during the study, they will not be deprived of routine treatment. Participants will receive either melatonin supplements or a placebo alongside their standard treatments.

## Harms

According to the findings of the meta-analysis conducted by Menczel Schrire et al. [31], even daily consumption of doses ≥ 10 mg of melatonin supplements did not have any significant complications except drowsiness, headache, and dizziness. If any complications arise with the melatonin supplementation, the intervention will be halted immediately, and appropriate medical actions will be taken.

## Study outcomes

The primary outcomes of the current trial are to reduce plasma levels of 2-AG, ghrelin, and the scores associated with hedonic eating. The secondary outcomes focus on dietary intake and body weight. Additionally, at the study's outset, participants will provide their medical history and demographic information, including age, gender, educational background, and marital status.

The intensity of hedonic eating in the first phase of the project and at the end of the study will be measured using the PFS questionnaire, which has been previously validated for the Iranian population [28]. The questionnaire comprises 15 items across three subscales: the "Food Available (FA)" subscale evaluates public thoughts about foods, the "Food Present (FP)" subscale assesses attraction to directly available food, and the "Food Tasted (FT)" subscale evaluates attraction to foods that have a favorable/ preferred taste. The likert scale will be used for scoring, ranging from 1 to 5, where a higher score indicates greater hedonic hunger [32].

Plasma levels of 2-arachidonoylglycerol and acylated ghrelin will be measured before and after the study. For biochemical analysis, A 5 cc blood sample will be collected from each participant after a 10–12 hour fast for biochemical analyses. The blood will be placed in EDTA-coated tubes, centrifuged at 3000 rpm at 4 °C for 10 minutes, and then it will be stored at -80 °C to determine the plasma levels of 2-AG and acylated ghrelin. ELISA kits provided by ZellBio GmbH, Veltinerweg, Germany, will be used to assess the plasma concentrations of 2-AG and acylated ghrelin.

For the evaluation of dietary intake, including total energy and macronutrient intake, participants will complete three 24-hour dietary recalls. This will include one weekend day and two weekdays, using a validated tool for dietary assessment. These recalls will be conducted through in person or phone interviews at the beginning of the study, after four weeks, and upon the completion of the intervention The Nutritionist IV software will be utilized to process and analyze the dietary information.

Body weight measurements will be taken using a Seca scale, accurate to 0.1 kg, at three key points: at the beginning of the study, after four weeks, and at the end of trial. During these measurements, participants will be wearing in light clothing.

Moreover, as a means of controlling for confounding factors, the validated short form of the Persian version of the International Physical Activity Questionnaire (IPAQ) will be used for the evaluation of physical activity at both the study's outset and its completion [33,34].

## Ethics approval and consent to participate

The study protocol is registered in the Iranian Registry of Clinical Trials (IRCT20080904001197N22) and adheres to the Declaration of Helsinki Guidelines. Approval has been obtained from the Ethics Committee of Tabriz University of Medical Sciences, Iran (IR.TBZMED.REC.1402.075) [S2 File]. Before conducting the trial, any alterations to the study protocol—such as changes in objectives, design, patient population, sample size, procedures, or administrative aspects—must receive confirmation from the relevant departments. This confirmation ensures that patient safety and potential benefits are considered. Written informed consent will be obtained from all participants. Withdrawal of consent is possible at any time without loss of benefit.

## Data management and monitoring

An appointed monitor for the clinical trial will periodically oversee the study's progression to protect the rights and welfare of the participants. This monitor will ensure

compliance with the study protocol, ethical norms, and legal standards, verify the presence of necessary documents, and confirm the accurate recording of data. A member of the research team will be responsible for verifying the data's coding, security, and storage procedures, as well as double-checking data entries and their corresponding values. Furthermore, participants will be followed for any negative reactions to melatonin intake. If such reactions are linked to melatonin use, affected individuals will be instructed to stop taking the supplement and immediately consult a specialist for additional care. In these instances, unblinding is allowed according to the criteria set by the Medical Ethics Committee.

## Auditing

This study will be overseen by the Ethics Committee of Tabriz University of Medical Sciences. The Committee will ensure the study's quality, validity, and adherence to ethical standards through at least two monitoring sessions. Additionally, a report will be submitted to the auditor every three months.

## Statistical analysis

Data will be analyzed statistically using SPSS 26.0 software, considering P values below 0.05 as indicative of statistical significance. The normality of data distribution will be checked using the Kolmogorov-Smirnov test, and an appropriate method will be applied to normalize variables that do not follow a normal distribution. Mean (standard deviation) and frequency (percentage) will be used to represent quantitative and qualitative variables, respectively. Independent sample t-tests will be used to evaluate differences in baseline means, while paired-sample t-tests will compare within-group differences. For variables assessed three times during the study (at the start, in the fourth week, and in the eighth week), a repeated measures ANOVA will be utilized. Additionally, Analysis of Covariance (ANCOVA) will assess the effects of the intervention between groups, adjusting for confounders such as baseline data and physical activity. In cases where dropout rates of 15% or more, an intention-to-treat (ITT) analysis will be performed. Missing data will be handled using multiple imputation methods.

## Confidentiality

Every participant will be assigned a unique identification code for data storage purposes. This code list will be accessible exclusively to researchers and safeguarded by the principal investigator during the trial. Participants' personal information will be kept confidential in all published reports.

## Plans to give access to the full protocol, participant-level data, and statistical code

The corresponding author will provide datasets, statistical code, and the full protocol upon reasonable request.

## Composition of the coordinating center and trial steering committee

The principal researcher coordinates the study, with regular communication among the study team and supervision by a steering committee. Additionally, the Ethics Committee at Tabriz University of Medical Sciences will monitor compliance with ethical guidelines, and the study will be revised or terminated if any ethical issues arise.

### Provisions for post-trial care

Melatonin or placebo supplements will be given for 8 weeks. Once the trial is completed, routine care will continue.

### Dissemination plans

The final results and data will be shared in upcoming publications.

### Trial status

The recruitment process for the first phase of the project began on January 8, 2024, and is expected to be completed by October 1, 2024.

## Discussion

Despite the importance of hedonic eating in the context of overweight and obesity, there are limited interventions aimed at reducing it. In a study by Beaumont et al., transcranial direct current stimulation targeting the dorsolateral prefrontal cortex—known for its role in controlling hedonic eating—did not impact hedonic appetite when administered at 2 mA for 20 minutes [35]. However, another study found that intragastric quinine-hydrochloride infusion at a dose of 10 μmol/kg effectively reduced hedonic food intake and plasma ghrelin levels in non-obese women [36]. Additionally, an education-based intervention developed by Mason and colleagues successfully decreased the hedonic drive to eat in adults with obesity after 6 months [37]. Furthermore, thylakoid supplementation, taken once daily for 3 months, reduced the urge for palatable food in women with overweight [38]. In addition, the 12-week use of behavioral weight loss programs in adults with overweight/obesity resulted in a reduction in hedonic hunger, and this reduction was associated with better weight loss [16,39].

Melatonin is a hormone with antioxidant and anti-inflammatory properties. It also has a significant role in glycemic homeostasis, modulation of white adipose tissue activity, lipid metabolism, and mitochondrial activity. In addition, melatonin increases brown adipose tissue volume and activity [19]. A meta-analysis in 2021 showed that taking daily doses of melatonin of ≤ 8 mg leads to weight loss. Nevertheless, it has been noted that the mechanisms underlying weight loss with melatonin supplementation are intricate and not completely comprehended [18]. In addition to the mechanisms mentioned, according to the results of animal studies, it appears that melatonin may also impact body weight by reducing hedonic eating. Some animal studies have shown a decrease in the serum level of ghrelin in mammals after melatonin supplementation [20–22]. In zebrafish, melatonin supplementation decreased the expression of the ghrelin hormone gene, and the CB1 receptor gene expression, as well as food intake [7,23]. Furthermore, norepinephrine, which is essential for the production of melatonin at night, down-regulated levels of some endocannabinoids, within the pineal gland of rats [24]. As previously mentioned, endocannabinoids and ghrelin are involved in the hedonic eating process. Therefore, melatonin supplementation may probably reduce food intake and hedonic eating by decreasing the levels of endocannabinoids, the expression of the ghrelin hormone gene, the level of ghrelin, and the expression of the CB1 receptor gene. In conclusion, it appears that conducting a clinical trial to examine the impact of melatonin supplementation on hedonic eating intensity and related biochemical biomarkers in overweight or obese individuals may be beneficial.

## Strengths and limitations

This pilot study's strengths lie in its use of a randomized, double-blind, placebo-controlled design to investigate the effects of melatonin supplementation on hedonic hunger intensity and related biochemical biomarkers in women with overweight/obesity, which is a novel approach. However, the research does have some limitations. Firstly, only two biochemical markers related to hedonic eating will be measured in this study. Secondly, despite employing a validated questionnaire for hedonic eating assessment, misclassification bias cannot be entirely eliminated. Lastly, despite accounting for potential changes in dietary intake or physical activity during statistical analysis, the participants' self-reported dietary intake and physical activity could still influence the results. We anticipate that this trial will provide a general idea of the effectiveness of melatonin intervention in managing hedonic eating.

## Supporting information

**S1 Checklist. SPIRIT 2013 checklist.**
(DOC)

**S1 File. Model consent form to participants.**
(DOCX)

**S2 File. Study protocol approved by the research ethics committee.**
(DOCX)

## Acknowledgments

This protocol is the second phase of a Ph.D. dissertation registered at Tabriz University of Medical Sciences, Tabriz, Iran. The authors would like to express their gratitude to Tehran Darou Company in Tehran, Iran for providing the necessary supplements and placebos.

## Author contributions

**Conceptualization:** Malihe Karamizadeh, Reza Mahdavi, Bahram Pourghassem Gargari.

**Investigation:** Malihe Karamizadeh.

**Methodology:** Malihe Karamizadeh, Reza Mahdavi, Bahram Pourghassem Gargari.

**Project administration:** Reza Mahdavi, Bahram Pourghassem Gargari.

**Supervision:** Marzieh Akbarzadeh, Reza Mahdavi, Bahram Pourghassem Gargari.

**Writing – original draft:** Malihe Karamizadeh, Azadeh Khalilitehrani, Neda Lotfi Yagin.

**Writing – review & editing:** Marzieh Akbarzadeh, Reza Mahdavi, Bahram Pourghassem Gargari.

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
