## [Decision Letter · Decision Letter 0]

14 Nov 2024

PONE-D-24-36320The effect of melatonin supplementation on the plasma levels of 2-arachidonoylglycerol, ghrelin and hedonic eating intensity in overweight/obese females: a study protocol for a pilot randomized controlled trialPLOS ONE

Dear Dr. Pourghassem Gargari,

Thank you for submitting your manuscript to PLOS ONE. After careful consideration, we feel that it has merit but does not fully meet PLOS ONE’s publication criteria as it currently stands. Therefore, we invite you to submit a revised version of the manuscript that addresses the points raised during the review process.

We look forward to receiving your revised manuscript.

Kind regards,

Marcello Iriti, Ph.D.

Academic Editor

PLOS ONE

**Journal Requirements:**

The funding for the trial received support from Tabriz University of Medical Sciences under grant number IR.TBZMED.REC.1402.075.

This protocol is the second phase of a Ph.D. dissertation registered at Tabriz University of Medical Sciences, Tabriz, Iran, with the registration number IR.TBZMED.REC.1402.075.

The funding for the trial received support from Tabriz University of Medical Sciences under grant number IR.TBZMED.REC.1402.075.

Reviewers' comments:

Reviewer's Responses to Questions

**Comments to the Author**

1. Does the manuscript provide a valid rationale for the proposed study, with clearly identified and justified research questions?

Reviewer #1: Yes

Reviewer #2: Yes

2. Is the protocol technically sound and planned in a manner that will lead to a meaningful outcome and allow testing the stated hypotheses?

Reviewer #1: Yes

Reviewer #2: Yes

3. Is the methodology feasible and described in sufficient detail to allow the work to be replicable?

Reviewer #1: Yes

Reviewer #2: Yes

4. Have the authors described where all data underlying the findings will be made available when the study is complete?

Reviewer #1: Yes

Reviewer #2: Yes

5. Is the manuscript presented in an intelligible fashion and written in standard English?

Reviewer #1: Yes

Reviewer #2: Yes

6. Review Comments to the Author

You may also provide optional suggestions and comments to authors that they might find helpful in planning their study.

**Reviewer #1: ** The authors intend to conduct a randomized, double-blinded, placebo-controlled study to investigate the effect of melatonin supplementation on plasma levels of 2-arachidonylglycerol and ghrelin, and intensity of hedonic eating. They will recruit forty-six women, randomly assigned to two arms: melatonin supplement or placebo with balanced design for 8 weeks. Data will be analyzed with ANCOVA.

1. Line 101. “46 subjects (23 in each group)”. The determined sample size seems quite arbitrary. What effect size can be detected by this sample size?

2. Line 114. “apparently healthy volunteers”. This sounds very subjective. Need to clarify how to define apparently healthy.

3. Line 148. “if any adverse events occur, unblinding is allowed”. Please clarify whether these participants will be included in the analysis.

4. Line 233. “log transformation will be applied to normalize variable”. As you do not know the distribution of variables yet, how do you know log-transformation will be appropriate?

**Reviewer #2: ** Dear Editor,

Thank you for inviting me to review this manuscript. I appreciate the opportunity to contribute to the peer review process for this study

General comments : The study makes a valuable contribution by investigating the effects of melatonin on hedonic eating in overweight/obese women, addressing an important gap in the literature. The introduction provides a strong background and context, with a clear and logical flow that lays a solid foundation for the research and leads smoothly to the study's main objective. The study design is robust, and the intervention plan is clearly defined. However, since the manuscript does not present any results, it limits the ability to fully evaluate the outcomes and implications of the research. Despite this, I believe this is an excellent study that could provide meaningful insights into the role of melatonin in eating behavior. However, there are some improvements that should be made prior to publishing.

Point 1 #In the introduction section, I suggest adding a line or two to briefly describe the PFS and explain its specific relevance to hedonic eating studies.

Point 2 # Could you clarify how the number of 46 participants was determined? It would be helpful to provide more details on how the sample size was calculated, and whether a power analysis was conducted. Also, please explain how the 15% attrition rate was estimated.

Point 2 # While you have have outlined a comprehensive set of inclusion and exclusion criteria, it’s still unclear how these have been verified during screening.

Point 3 #The block size is mentioned as two. It might be helpful to confirm that this approach maintains the required balance across groups, as the small block size may not provide sufficient flexibility to ensure even distribution, particularly toward the end of the randomization process.

Point 4# It’s good that one author (RM) created the randomization sequence without being involved in assigning participants. This could be even stronger if it’s mentioned whether they are also blinded to any participant details or baseline data to reduce bias further.

Point 5 # Please specify the software used for randomization.

Point 6# Specify the purpose of bi-weekly follow-up calls.

Point 7# Please clarify if there’s a standard method for counting and recording any unused tablets at each checkpoint to help ensure accurate tracking of compliance.

7. PLOS authors have the option to publish the peer review history of their article (what does this mean? ). If published, this will include your full peer review and any attached files.

**Do you want your identity to be public for this peer review?** For information about this choice, including consent withdrawal, please see our Privacy Policy .

Reviewer #1: No

Reviewer #2: No

---

## [Author Response · Author response to Decision Letter 1]

15 Jan 2025

Dear Dr. Marcello Iriti,

Thank you very much for your comments on our manuscript entitled as: The effect of melatonin supplementation on the plasma levels of 2-arachidonoylglycerol, ghrelin and hedonic eating intensity in overweight/obese females: a study protocol for a pilot randomized controlled trial; Manuscript ID: PONE-D-24-36320. We really appreciate your time and consideration. The following are our point-by-point responses to the comments. We have highlighted all changes in the new format of the manuscript.

REVIEWER REPORT(S):

Referee: 1

Comments to the Author

-) Line 101. “46 subjects (23 in each group)”. The determined sample size seems quite arbitrary. What effect size can be detected by this sample size?

As noted in p.6, lines 121-127 currently, no human studies have investigated the use of melatonin supplementation on hedonic eating. The proposed study is a pilot project; its main aim is to establish data for sample size calculations for a larger trial. Various studies recommend a total sample size of at least 24 participants for a two-arm trial. Considering the sample size of previous studies on melatonin supplementation in overweight and obese Iranian women (Mohammadi et al, https://doi.org/10.1155/2021/3502325, and Mesri Alamdari et al, http://dx.doi.org/ 10.1055/s-0034-1384587) and to increase the accuracy and power of the study, in this clinical trial, we plan to enroll 40 participants, which, with a 15% attrition rate, will increase to 46 subjects (23 in each group).

-) Line 114. “apparently healthy volunteers”. This sounds very subjective. Need to clarify how to define apparently healthy.

Apparently healthy volunteers are individuals who have not been diagnosed with any medical conditions and are not taking any medications (p.7, lines: 140-141).

-) Line 148. “if any adverse events occur, unblinding is allowed”. Please clarify whether these participants will be included in the analysis.

If dropout rates reach 15% or higher due to adverse effects, we will conduct an intention-to-treat (ITT) analysis (p.9, lines: 175-176).

-) Line 233.“log transformation will be applied to normalize variable”. As you do not know the distribution of variables yet, how do you know log-transformation will be appropriate?

It was changed to: The normality of data distribution will be checked using the Kolmogorov-Smirnov test, and an appropriate method will be applied to normalize variables that do not follow a normal distribution (p.13, lines: 272-274).

Referee: 2

Comments to the Author

-) Suggestion adding a line or two to describe the PFS briefly In the introduction section

It was added to manuscript (p.5, lines: 92-94).

-) Could you clarify how the number of 46 participants was determined? It would be helpful to provide more details on how the sample size was calculated, and whether a power analysis was conducted. Also, please explain how the 15% attrition rate was estimated.

As noted in p.6, lines 121-127 currently, no human studies have investigated the use of melatonin supplementation on hedonic eating. The proposed study is a pilot project; its main aim is to establish data for sample size calculations for a larger trial. Various studies recommend a total sample size of at least 24 participants for a two-arm trial. Considering the sample size of previous studies on melatonin supplementation in overweight and obese Iranian women (Mohammadi et al, https://doi.org/10.1155/2021/3502325, and Mesri Alamdari et al, http://dx.doi.org/ 10.1055/s-0034-1384587) and to increase the accuracy and power of the study, in this clinical trial, we plan to enroll 40 participants, which, with a 15% attrition rate, will increase to 46 subjects (23 in each group). Regarding the attrition rate, the dropout rate can be determined using historical data from similar studies or pilot studies. A common rule of thumb is to assume a 10-20% dropout rate (Patel et al, DOI:10.55489/njcm.150620243815). Studies conducted on melatonin supplementation in Iran (such as Mohammadi et al, https://doi.org/10.1155/2021/3502325, Mesri Alamdari et al, http://dx.doi.org/ 10.1055/s-0034-1384587, Jamilian et al, https://doi.org/10.3389/fendo.2019.00273, Shabani et al, https://doi.org/10.1016/j.jad.2019.02.066, Bahrami et al, https://doi.org/10.1108/NFS-012019-0018, Bahrami et al, https://doi.org/10.1016/j.ctim.2020.102452, Pakravan et al, https://doi.org/10.4103/2277-9175.204593) found a maximum rate of 12.5%. Therefore, we have decided to consider the attrition rate to be 15%.

-) While you have outlined a comprehensive set of inclusion and exclusion criteria, it’s still unclear how these have been verified during screening.

As mentioned in the lines 129-131, the present pilot study is the second phase of a Ph.D. dissertation. The first phase of the project is cross-sectional, and approximately 400 overweight or obese females will be recruited. The mentioned inclusion and exclusion criteria will be assessed in the first phase of the project (p.7, lines: 151-153).

-) The block size is mentioned as two. It might be helpful to confirm that this approach maintains the required balance across groups, as the small block size may not provide sufficient flexibility to ensure even distribution, particularly toward the end of the randomization process.

We will change the block size to four (p.8, lines 164-165).

-) It’s good that one author (RM) created the randomization sequence without being involved in assigning participants. This could be even stronger if it’s mentioned whether they are also blinded to any participant details or baseline data to reduce bias further.

We applied this suggestion (p.9, line 180).

-) Please specify the software used for randomization.

Randomization is conducted using the Random Allocation Software (RAS). It is mentioned in p.8, lines 163-164.

-) Specify the purpose of bi-weekly follow-up calls.

We changed this phrase to: weekly follow-up calls to improve response rates and fewer dropouts (p.9, lines 189-190).

-) Please clarify if there’s a standard method for counting and recording any unused tablets at each checkpoint to help ensure accurate tracking of compliance.

The total intake of tablets for each subject is calculated by subtracting the number of tablets unused from the number of tablets dispensed (p.9, lines 195-196).

With Regards,

Reza Mahdavi

Nutrition Research Center, Department of Biochemistry and Diet Therapy, Faculty of Nutrition and Food Sciences, Tabriz University of Medical Sciences, Tabriz, Iran

Email: mahdavir@tbzmed.ac.ir

Bahram Pourghassem Gargari

Nutrition Research Center, Department of Biochemistry and Diet Therapy, Faculty of Nutrition and Food Sciences, Tabriz University of Medical Sciences, Tabriz, Iran

Email: pourghassemb@tbzmed.ac.ir

---

## [Decision Letter · Decision Letter 1]

30 Jan 2025

The effect of melatonin supplementation on the plasma levels of 2-arachidonoylglycerol, ghrelin and hedonic eating intensity in overweight/obese females: a study protocol for a pilot randomized controlled trial

PONE-D-24-36320R1

Dear Dr. Pourghassem Gargari,

We’re pleased to inform you that your manuscript has been judged scientifically suitable for publication and will be formally accepted for publication once it meets all outstanding technical requirements.

Kind regards,

Marcello Iriti, Ph.D.

Academic Editor

PLOS ONE

Additional Editor Comments (optional):

Reviewers' comments:

Reviewer's Responses to Questions

**Comments to the Author**

1. Does the manuscript provide a valid rationale for the proposed study, with clearly identified and justified research questions?

Reviewer #1: Yes

Reviewer #2: Yes

2. Is the protocol technically sound and planned in a manner that will lead to a meaningful outcome and allow testing the stated hypotheses?

Reviewer #1: Yes

Reviewer #2: Yes

3. Is the methodology feasible and described in sufficient detail to allow the work to be replicable?

Reviewer #1: Yes

Reviewer #2: Yes

4. Have the authors described where all data underlying the findings will be made available when the study is complete?

Reviewer #1: Yes

Reviewer #2: Yes

5. Is the manuscript presented in an intelligible fashion and written in standard English?

Reviewer #1: Yes

Reviewer #2: Yes

6. Review Comments to the Author

You may also provide optional suggestions and comments to authors that they might find helpful in planning their study.

Reviewer #1: Thank you for addressing the raised comments and made the edits accordingly. This reviewer has no further concern.

Reviewer #2: Dear Editor,

Thank you again for inviting me to review the revised manuscript for your esteemed journal.

The manuscript, titled « The effect of melatonin supplementation on the plasma levels of 2- arachidonoylglycerol, ghrelin and hedonic eating intensity in overweight/obese females: a study protocol for a pilot randomized controlled trial » (Manuscript ID: PONE-D-24-36320R1)," by Malihe Karamizadeh et al., has been appropriately revised. The authors have adequately addressed all the concerns raised. I find the revised manuscript to be well-prepared and scientifically robust. Therefore, I believe it merits publication in PLOS One.

Dear Authors, I have no further comments. Thank you for your efforts in addressing the previous concerns.

7. PLOS authors have the option to publish the peer review history of their article (what does this mean? ). If published, this will include your full peer review and any attached files.

**Do you want your identity to be public for this peer review?** For information about this choice, including consent withdrawal, please see our Privacy Policy .

Reviewer #1: No

Reviewer #2: No

---

## [Editor Report · Acceptance letter]

PONE-D-24-36320R1

PLOS ONE

Dear Dr. Pourghassem Gargari,

I'm pleased to inform you that your manuscript has been deemed suitable for publication in PLOS ONE. Congratulations! Your manuscript is now being handed over to our production team.

Kind regards,

on behalf of

Prof. Marcello Iriti

Academic Editor

PLOS ONE